Current freezing and thawing scenarios employed by North Atlantic fisheries: their potential role in Newfoundland and Labrador’s northern cod (Gadus morhua) fishery

Brown Pete
Dave Deepika deepika.dave@mi.mun.ca
Centre for Aquaculture and Seafood Development, Fisheries and Marine Institute of Memorial University, Memorial University of Newfoundland , St. John’s , NL , Canada
Bilbao-Sainz Cristina
Electronic publication date: 2021 Dec 10
Publication date: 2021
Volume: 9
Electronic Location ID: e12526
Received 2021 Jul 16; Accepted 2021 Oct 30
Copyright: ©2021 Brown and Dave
Copyright year: 2021
Copyright holder: Brown and Dave
License: This is an open access article distributed under the terms of the Creative Commons Attribution License, which permits unrestricted use, distribution, reproduction and adaptation in any medium and for any purpose provided that it is properly attributed. For attribution, the original author(s), title, publication source (PeerJ) and either DOI or URL of the article must be cited.
License URL: https://creativecommons.org/licenses/by/4.0/

Keywords: Fisheries, Seafood, Freezing, Thawing, Spoilage, Preservation, Global value chains, Northern cod

Funding: The authors received no funding for this work.

==============================
Seafood is very perishable and can quickly spoil due to three mechanisms: autolysis, microbial degradation, and oxidation. Primary commercial sectors within the North Atlantic fisheries include demersal, pelagic, and shellfish fisheries. The preservation techniques employed across each sector can be relatively consistent; however, some key differences exist across species and regions to maintain product freshness. Freezing has long been employed as a preservation technique to maintain product quality for extended periods. Freezing allows seafood to be held until demand improves and shipped long distances using lower-cost ground transportation while maintaining organoleptic properties and product quality. Thawing is the opposite of freezing and can be applied before additional processing or the final sale point. However, all preservation techniques have limitations, and a properly frozen and thawed fish will still suffer from drip loss. This review summarizes the general introduction of spoilage and seafood spoilage mechanisms and the latest preservation techniques in the seafood industry, focusing on freezing and thawing processes and technologies. This review also considers the concept of global value chains (GVC) and the points to freeze and thaw seafood along the GVC to improve its quality with the intention of helping Newfoundland and Labrador’s emerging Northern cod (Gadus morhua) fisheries enhance product quality, meet market demands and increase stakeholder value.

Introduction

Seafood is highly perishable and very susceptible to spoilage. Ghaly et al. (2010) reported that 30% of seafood is lost annually due to spoilage. Therefore, food preservation, quality, and storage improvements are essential to increasing value and expanding markets for seafood harvesters and processors. Three primary sectors within the North Atlantic fisheries include demersal, pelagic, and shellfish fisheries. The preservation techniques employed across each fishery can be relatively consistent with slight variations across species and regions.

Spoilage is the result of three mechanisms: autolysis, microbial degradation, and oxidation. Preservation techniques allow slowing down spoilage mechanisms and help to extend the shelf-life of food products (Vaclavik & Christian, 2014). Typical preservation methods commonly used by seafood processors include drying, salting, smoking, pickling, canning, heating, fermenting, chilling, freezing, high-pressure processing (HPP), and modified atmospheric packaging (MAP) (Ghaly et al., 2010; Lauzon et al., 2010; Sigurgisladottir et al., 2000; Vaclavik & Christian, 2014). For centuries, freezing has been employed as a preservation technique to maintain product quality for extended periods (Kaloyereas, 1950). Freezing allows seafood to be stored until demand improves, transported long distances at lower costs to markets, and maintain organoleptic properties and quality before secondary processing or final sale. Freezing has been consistently used in commercial applications since Charles Birdseye invented the double belt contact freezer in 1928 (Persson & Löndahl, 1993). Since then, quick-freezing is still the primary process for preserving many seafood products for a few weeks to several months while protecting their original sensory properties (Backi, 2018; Hedges, 2002; Torry Research Station, 1977).

Freezing is the subtraction of heat to lower the product’s temperature sufficiently that most of the intra- and extracellular water undergoes a phase change from a liquid to a solid (Backi, 2015; Haugland, 2002; Reid, 1993). Freezing techniques employed to slow down autolytic and microbiological degradation and preserve seafood include freezing, frozen storage and thawing (Hanenian & Mittal, 2004). However, holding products in cold storage for extended periods can still lead to the oxidation of lipids and other fats and impact a product’s sensory properties and safety aspects (Frankel, 1985; Papuc et al., 2017; Secci & Parisi, 2016). Lowering the cold storage temperature, reducing the time held in cold storage, reducing temperature fluctuations, or combining freezing with other preservation methods such as high-pressure processing (HPP), smoking, MAP, and improved packaging materials can help mitigate oxidation and enhance the quality and yield of seafood (Baygar & Alparslan, 2015; Dang et al., 2018b; Dawson, Al-Jeddawi & Remington, 2018; Fuentes et al., 2013; Lauzon et al., 2010; Nakazawa & Okazaki, 2020; Tironi, de Lamballerie & Le-Bail, 2010).

Seafood products must be thawed before secondary processing or the final point of sale to high-end clients. Thawing is the opposite of freezing; it is the addition of heat to raise the product’s temperature and enables the subsequent phase change of the intra- and extracellular water from a solid to a liquid (Backi, 2015; Haugland, 2002; Reid, 1993). After thawing, the product again has a shortened shelf life and should be processed or sold quickly. A hybrid supply chain where the frozen product undergoes secondary processing and is refrozen, referred to as “twice frozen”, is commonly used for Alaskan Pollock (Gadus chalcogrammus) to extend its shelf life (Hedges, 2002). However, all preservation techniques have their limitations, and a properly frozen and thawed fish can lose 5% of its weight in drip loss (Dinçer et al., 2009).

To the best of our knowledge, this paper is the first of its kind comprehensive review that attempts to: introduce general spoilage and seafood spoilage mechanisms, define preservation techniques employed by the seafood industry to extend shelf-life, and summarize the freezing and thawing processes and technologies used to improve the quality of North Atlantic seafood. It also discusses the Global Value Chains (GVC) for North Atlantic seafood and utilizing freezing and thawing processes along the GVC to improve the quality and value for seafood products; specifically, demersal, pelagic and shellfish species. Moreover, this paper synthesizes applying these technologies to enhance product quality, meet market demands and increase stakeholder value for the re-emerging Northern Cod (Gadus morhua) fishery in Newfoundland and Labrador, Canada. However, the authors have since identified an article with a similar scope (Svendsen et al., 2022).

The intended audience for this review paper includes engineers, researchers, food scientists, and students working in the seafood preservation sector. This paper is also expected to appeal to industry players, specifically food processors looking for guidance to improve product quality and shelf life.

Survey Method

To ensure an unbiased review of the literature, the authors ran searches using the Memorial University of Newfoundland Library, WorldCat, Web of Knowledge and Google Scholar databases. Keywords used for these searches included: spoilage, preservation, freezing, thawing, seafood and value chain, and keywords were used individually and in combination. Papers identified by each search were initially scanned to determine their appropriateness to the subject, and all relevant articles were read in detail. To increase the breadth of this review, the authors also examined relevant papers referenced by the papers identified during the initial searches.

Postmortem degradation of seafood

Definition of spoilage

Spoilage is the deterioration of food material due to biological activities (Biodeterioration, 2006), and fish are amongst the most perishable food products. Even with regular refrigerated storage conditions or super-chilling, oxidative, enzymatic, and microbiological spoilage limits fish and fish products’ shelf-life (Lauzon et al., 2010; Olafsdóttir et al., 2006; Roiha et al., 2018a). Chemical, microbial, and enzymatic actions are responsible for the biological activities that cause food spoilage (Ghaly et al., 2010; Lakshmanan, 2000; Roiha et al., 2018a). There are three primary causes for spoilage in seafood (Huss et al., 1995; Surette, Gill & LeBlanc, 1988; Tull, 1996), including autolytic degradation, bacterial spoilage, and oxidation. Autolytic degradation refers to the self-digestion of the flesh. This degradation includes the natural decay caused by enzymatic changes within the muscle. Bacterial spoilage refers to the contamination of meat by microorganisms. In comparison, oxidation refers to the breakdown of lipids in the flesh due to heat, light, and oxygen. Therefore, proper handling and processing protocols must be employed across the supply chain to minimize spoilage, extend the shelf life of seafood and supply a high-quality product that is safe for consumers (Backi, 2018).

Autolytic spoilage

Huss et al. (1995) define autolytic degradation as self-digestion. Autolysis results from enzymatic changes in the muscle caused by the loss of respiration and homeostasis and digestive enzymes in the gut breaking down the flesh (Backi, 2015; Surette, Gill & LeBlanc, 1988). Autolysis is responsible for textural changes to the flesh while in storage (Benjakul & Bauer, 2000; Haugland, 2002) and reduced freshness (i.e., sensory impairment) with minor microbiological impacts (Huss et al., 1995).

Glycolysis is the primary enzymatic change occurring in the fish muscle after the death of the animal. The gradual hydrolysis of glycogen into lactic acid instead of carbon dioxide and water due to the lack of oxygen and results in textural changes as the animal goes through rigour-mortis (rigour) (Lakshmanan, 2000). Glycolysis can be reduced by minimizing the animal’s stress (Ghaly et al., 2010).

Nucleotide degradation typically follows glycolysis and is responsible for flavour changes in seafood and meat products (Alasalvar et al., 2002; Gram & Dalgaard, 2002; Lakshmanan, 2000; Surette, Gill & LeBlanc, 1988). Adenosine diphosphate (ATP) degrades to adenosine monophosphate (AMP) and then to inosine monophosphate (IMP) soon after the animal’s death (Alasalvar et al., 2002; Surette, Gill & LeBlanc, 1988). Surette, Gill & LeBlanc (1988) further reported the subsequent production of inosine (INO), hypoxanthine (Hx), xanthine (Xa) and uric acid (URATE) was rate-limited based on the freshness factor (Ki), calculated according to Eq. (1) (Karube et al., 1984). (1) Ki=Hx+InoIMP+Ino+Hx×100

where: Ki = freshness factor

Microbial spoilage

Food spoilage is a significant concern as the world loses 25% of the food supply and 30% of landed fish due to microbial activities (Ghaly et al., 2010). Microbial degradation usually follows autolytic spoilage and further deteriorates the sensory properties (taste, flavour, odour, colour, texture, and juiciness). Furthermore, microbial degradation of the muscle makes it unsuitable for human consumption (Ghaly et al., 2010; Gram & Dalgaard, 2002). Freshness, the opposite of spoilage, is an essential aspect of food products that require preservation (Cheng et al., 2014).

Seafood is at its peak freshness immediately after death, and seafood spoilage begins as soon as the animal dies (Huss et al., 1974; Lakshmanan, 2000). Therefore, fish harvesters and processors need to understand the effects of spoilage and antemortem and postmortem effects on the muscle (Ashie, Smith & Simpson, 1996; Nathanailides et al., 2011; Warrier, Gore & Kumta, 1975). The primary specific spoilage bacteria (SSBs) of concern for cold-water marine fish are Pseudomonas spp., Shewanella putrefaciens, Photobacterium phosphoreum, and Proteus spp. (De Alba et al., 2019; Gram & Dalgaard, 2002; Roiha et al., 2018b). As they produce H2S under aerobic conditions, these bacteria break down the animal’s flesh by secreting enzymes as they multiply (Castell, 1954; Gram & Huss, 1996) and lead to microbial and enzymatic spoilage.

Under suitable environmental conditions, seafood becomes a source of nutrients that can support the growth of a wide range of bacteria (Castell & Anderson, 1948; Gram & Dalgaard, 2002; Lakshmanan, 2000; Seremeti, 2007). Furthermore, these bacteria remain active under temperatures below 0 °C; however, their activity is reduced with every degree drop in temperature (Castell, 1954; Ghaly et al., 2010; Huss et al., 1995). Table 1 presents the decrease in fish products’ expected shelf life with increasing temperature (Huss et al., 1995).

Table 1 Time-temperature effect on spoilage for fish products.

Source: Huss et al., 1995.

Shelf life in days of fish products stored in ice (0 °C)	Shelf life at chill temperatures (days)	
	5 °C	10 °C	15 °C	
6	2.7	1.5	1	
10	4.4	2.5	1.6	
14	6.2	3.5	2.2	
18	8.0	4.5	2.9	

Oxidation spoilage

As seafood is held in cold storage, composition changes result in lipid and protein oxidation, protein degradation, and the loss of other vital molecules (Ghaly et al., 2010; Papuc et al., 2017). Oxidation spoilage is the reaction between oxygen and unsaturated lipids. It produces a wide range of unwanted compounds responsible for rancidity, undesirable flavours and odours, and potential toxicity of the flesh (Badii & Howell, 2002; Frankel, 1985; Papuc et al., 2017; Secci & Parisi, 2016). Lipid oxidation typically occurs when a product is held for too long in cold storage or at high temperatures (Castell, 1954; Dawson, Al-Jeddawi & Remington, 2018; Indergård et al., 2014).

The three stages of lipid oxidation are initiation, propagation, and termination (Ghaly et al., 2010; Kolanowski, Jaworska & Weißbrodt, 2007). During the initiation phase, lipid-free radicals are formed by adding heat and oxygen to form peroxyl radicals. During the propagation phase, hydroperoxides are created when the peroxyl radicals react with other lipids. Finally, secondary oxidation products such as hydrocarbons, aldehydes and ketones are formed during the termination phase, and unsaturated aldehydes and ketones are oxidized further to produce volatile compounds. These changes result in off-flavours and signal that the product is no longer edible (Ghaly et al., 2010; Kolanowski, Jaworska & Weißbrodt, 2007; Lorentzen et al., 2019).

Oxidation typically occurs when the product is held for too long in cold storage or at high temperatures (Castell, 1954; Dawson, Al-Jeddawi & Remington, 2018; Indergård et al., 2014). Oxidation usually affects frozen seafood and results in a typical cold storage flavour, off-colours and protein breakdown (Badii & Howell, 2002; Bøknæs et al., 2000). Off-flavours typically increase with time and temperature during frozen storage and result from the formation of hept-cis-4-enal, hept-trans-2-enal and hept-trans-2, cis-4-dienal (McGill et al., 1974; McGill, Hardy & Gunstone, 1977). Badii & Howell (2002) reported that oxidation of lean flesh such as cod resulted in texture changes. Nakazawa & Okazaki (2020) and Tolstorebrov, Eikevik & Bantle (2016) reported that rapid freezing and cold storage at −30 °C helped minimize protein denaturation. Therefore, optimizing the freeze time and temperature of frozen storage for seafood is essential for maintaining quality, especially for fish with higher lipid content.

Cod spoilage

Understanding the mechanisms that cause spoilage in cod is essential for implementing preservation techniques and optimizing value. In cod, autolytic degradation occurs first, followed by microbiological spoilage. Ghaly et al. (2010), Gram & Dalgaard (2002) and Huss et al. (1995) discussed the significant contribution of microbial growth and autolytic degradation responsible for fish spoilage. Therefore, it is essential to minimize spoilage while optimizing the value of the resources. Table 2 summarizes the four phases of eating quality for cod stored on ice (Huss et al., 1995).

Table 2 Phases of cod spoilage.

Source: Huss et al., 1995.

Phase	Description	Time	Characteristics	
I	Fresh	0–2 days	Sweet, seaweedy and delicate taste; can be slightly metallic.	
II	Neutral	2–6 days	Loss of characteristic taste and odour; no off-flavours. The texture is still pleasant.	
III	Stale	6–12 days	Spoilage increases during this phase and produces a range of volatile, unpleasant odours. Trimethylamine (TMA) production generates the ”fishy” smell.	
IV	Spoiled	12–15 days	Spoiled and putrid	

Seafood preservation techniques

Slowing down spoilage mechanisms is the objective of food preservation techniques (Vaclavik & Christian, 2014). Kaloyereas (1950) provided a detailed history of food preservation techniques and their adoption by early civilizations who learned the benefits of preserving foodstuffs, likely by accident, and discussed the impacts science has had on the progression of preservation methods. Conventional preservation techniques used in industry include salting, drying, smoking, pickling, canning, heating, fermenting, chilling, and freezing (Ghaly et al., 2010; Lauzon et al., 2010; Vaclavik & Christian, 2014). However, many of these techniques may change the product’s organoleptic properties, such as taste, flavour, odour, texture, and juiciness. The following sections of this review paper will focus on freezing and thawing as preservation methods; if employed correctly, allow products to retain their fresh taste and texture (Kolbe & Kramer, 2007).

Freezing and freezing techniques

Retaining the quality of fish is the primary focus of preservation techniques such as freezing. Freezing is the most common preservation method for many seafood products and has been used for thousands of years to help maintain high product quality (Alizadeh et al., 2007; Gonçalves, Nielsen & Jessen, 2012). Freezing, as a preservative, typically includes three operations: freezing, frozen storage and thawing (Hanenian & Mittal, 2004). Freezing fish lowers the product’s temperature and slows down spoilage, so when the product is thawed, it is virtually identical to fresh fish (Torry Research Station 1977). However, the ice crystals formed during the process can severely damage the product, so the freezing kinetics should be designed to optimize the ice crystals’ size, shape, and distribution (Otero et al., 2016).

Freezing entails two linked processes: lowering the temperature sufficiently to remove the sensible and latent heat from a product and subsequent phase change of the intra- and extracellular water from a liquid to solid (Backi, 2015; Haugland, 2002; Reid, 1993; Singh & Heldman, 2009). Freezing consists of four phases, precooling, nucleation, phase transition and tempering (Otero et al., 2016). Eq. (2) describes the thermodynamic relationship for freezing (Backi, 2018). (2) dh=cTTi−Tf+Lf

Where: dh = the heat energy removed from the fish

c(T) = the specific heat capacity (temperature dependent)

Ti = the initial temperature of the fish

Tf = the final, or desired, temperature of the fish

Lf = the latent heat of fusion

In this case, the latent heat of fusion is the amount of energy that must be subtracted to undergo the phase change (Vaclavik & Christian, 2014).

Freezing minimizes water activity and slows down the deterioration caused by enzymes and microorganisms by reducing the product’s temperature below their active range (Connell, 1980; Svendsen et al., 2022) and its use as a preservative was first attributed to the Chinese around 100 BC (Kaloyereas, 1950). More recently, William Cullen, Jacob Perkins, Ferdinand Carré, Carl von Linde, and Clarence Birdseye developed and improved the freezing technology currently used in many modern seafood processing plants (Appel, 1990; Kaloyereas, 1950; Persson & Löndahl, 1993).

Freezing also enables processors to optimize processing, transporting, marketing, and retailing requirements by extending product freshness and quality for a few weeks to several months (Backi, 2018; Fuentes et al., 2013). Cold storage may also permit the use of slower, less carbon-intensive transportation methods to ship products to lower-cost secondary processors or markets (Ziegler et al., 2013).

However, freezing techniques have limitations. A typical, properly frozen and thawed fish can lose 5% of its weight through drip loss (Dinçer et al., 2009). Once frozen, seafood must be held in cold storage at -18 °C or below to preserve its freshness (FAO & WHO 2020). However, Nakazawa & Okazaki (2020) reported that even at the temperatures that most commercial ethylene-glycol freezers are designed to operate at (−20 °C to −25 °C), protein denaturation for many cold-water species may still occur due to the presence of psychrophiles. Therefore, colder storage temperatures of −30 °C to −40 °C are recommended to maintain quality. Furthermore, if this cold chain is not maintained, thawing and subsequent refreezing will result in ice crystal formation and damage the flesh (Dawson, Al-Jeddawi & Remington, 2018; Fuentes et al., 2013; Gonçalves, Nielsen & Jessen, 2012; Washburn et al., 2017). This thermal abuse results in additional drip loss and will further degrade the organoleptic and sensory properties such as taste, flavour, odour, colour, texture, and juiciness of the fish (Garthwaite, 1997; Karthikeyan et al., 2015; Rehbein, 2002). It is also crucial for the processors to replace their stock on a first-in, first-out basis to help prevent oxidative spoilage by minimizing the cold storage time. This spoilage is significant when species contain high lipid content (Ghaly et al., 2010; Kolanowski, Jaworska & Weißbrodt, 2007; Lorentzen et al., 2019).

Studies have shown that slower freezing rates typically result in the formation of larger ice crystals and severe damage to the cellular structure, osmotic water loss, denaturation of protein and mechanical damage to the fillet (Alizadeh et al., 2007; Benjakul & Bauer, 2000; Chevalier et al., 2000). Small amounts of cellular damage are hardly noticeable as the product will reabsorb the lost water during the thawing process. However, Nakazawa & Okazaki (2020) reported that improper freezing and cold storage would result in significant protein denaturation and cellular damage such as gaps and drip loss. Rapid freezing freezes the product more quickly, reduces ice crystal size, minimizes damage to the flesh, captures the product’s quality and freshness and holds it until the product is thawed (Backi, 2018; Benjakul & Visessanguan, 2010; Chevalier et al., 2000; Otero et al., 2016; Rehbein, 2002). Rapid freezing techniques employed by seafood processors worldwide include blast freezing, plate freezing, impingement tunnel freezing, immersion freezing, and cryogenic freezing.

Commercial freezers typically fall into two categories, (a) indirect and (b) direct contact (Fig. 1) Direct contact systems (i.e., air blast and immersion freezers) put the product in direct contact with the freezing medium. In contrast, indirect systems (i.e., plate freezers) operate by placing a barrier between the product and cooling medium (Singh & Heldman, 2009). Both direct and indirect contact systems can apply to batch (where the product remains stationary) or continuous (where the product is conveyed through the freezer) configurations (Fig. 2) (Seremeti, 2007). Newfoundland and Labrador seafood processors employ these commercial freezing systems (slowest to fastest): still, blast, plate, impingement tunnel, and immersion freezing. Higher speed processes, such as cryogenic freezing, are less common in Newfoundland and Labrador, Canada, due to the lack of availability and cost of the gasses required to operate them. Several novel freezing methods that have not found commercial success include ultralow temperature freezing, ultrarapid freezing, dehydrofreezing, ultrahigh pressure and ultrasound (Li & Sun, 2002; Wu et al., 2017). Once frozen, fish are held in cold storage at −18 °C or lower, awaiting further distribution (FAO & WHO, 2020).

Figure 1 Schematic views of freezing: (A) Indirect contact freezing systems, and (B) direct contact freezing systems.

Source: Singh & Heldman, 2009.

Figure 2 Schematic depiction of freezers: (A) Typical batch air blast freezer, and (B) typical continuous tunnel freezer.

Still freezing

Walk-in and other still air freezers (Fig. 3A) function by holding the product in an insulated, cold air chamber at −5 to −30 °C until the product freezes using natural circulation and limited circulation from evaporator fans (Dempsey & Bansal, 2012; Kolbe & Kramer, 2007). The heat transfer mechanism is convection using air as the cooling medium (Laguerre & Flick, 2004). Air is a poor conductor of heat, so cooling rates tend to be slower in still air freezers; consequently results in the formation of larger, primarily extracellular ice crystals, more significant cellular damage and greater drip loss (Dempsey & Bansal, 2012; Jessen, Nielsen & Larsen, 2014; Kolbe & Kramer, 2007; Singh & Heldman, 2009; Suh et al., 2017). Kolbe & Kramer (2007) reported a typical heat transfer coefficient U = 5.7 W m−2 K−1, and Fellows (2016) reported a freezing rate of two mm h−1 for still air freezers. On the positive side, still freezers are relatively inexpensive to purchase and operate; therefore, typically used in commercial applications for cold storage or transportation of frozen products (Boonsumrej et al., 2007; Jessen, Nielsen & Larsen, 2014; Norsworthy, 2015).

Figure 3 Examples of freezing systems: (A) Still air, (B) air blast, (C) tunnel, (D) horizontal plate, (E) cryogenic, (F) brine immersion, (G) vertical plate, and (H) spiral.

Photo and drawing credits: Pete Brown.

Blast freezing

Air-blast freezers (Fig. 3B) are commonly used in commercial settings, especially land-based plants, to freeze seafood (Backi, 2018; Dempsey & Bansal, 2012; Jessen, Nielsen & Larsen, 2014). Blast freezers are considered quick freezers and can reduce the product’s temperature through the critical zone (0 °C to −5 °C) in less than 5 to 10 h (Fellows, 2016; Jessen, Nielsen & Larsen, 2014; Suh et al., 2017). Typical refrigerant temperatures range between −35 °C to −52 °C and air temperatures between −26 °C to −40 °C (Alizadeh et al., 2007; Boonsumrej et al., 2007; Cheng et al., 2014; Dempsey & Bansal, 2012). Garthwaite (1997) reported that air speeds around 5 m s−1 are economical for batch processes; however, 10–15 m s−1 are recommended air speeds for continuous processes to reduce the contact time with the freezing media. Fellows (2016) and Johnston et al. (1994) reported blast freezing rates of 2 to 5 mm h−1. Kolbe & Kramer (2007) stated a typical heat transfer coefficient U = 22.7 W m−2 K−1, confirming the relative increase in cooling rate of forced convection vs. natural convection.

Tunnel (Fig. 3C) and spiral freezers (Fig. 3H) are examples of continuous blast freezers. These freezers operate the same as batch-style blast freezers, except there is a constant product transfer through the chamber and direct application of cold air on the foodstuff (Backi, 2015). This direct cooling produces a higher typical heat transfer coefficient U = 56.8 W m−2 K−1 and faster relative freezing rates up to 30 mm h−1 (Fellows, 2016; Johnston et al., 1994; Kolbe & Kramer, 2007). Air-impingement freezers are examples of a straight-belt freezer where the product is cooled from the top and bottom with high-velocity cold air. These systems cost less than cryogenic freezers and have heat transfer coefficients approaching 350 W m−2 K−1 (Dempsey & Bansal, 2012; Fellows, 2016).

In contrast, air blast freezers need regular downtime for defrosting, especially with continuous processes (Garthwaite, 1997). They are also more expensive to operate because their large fans require considerable electricity to operate and cause a significant impact on the cooling system’s total heat load (Dempsey & Bansal, 2012). Moreover, there is a potential for dehydration leading to lipid oxidation and freezer burn in unwrapped and unglazed products, especially at higher airflow rates (Dempsey & Bansal, 2012; Jessen, Nielsen & Larsen, 2014; Seremeti, 2007).

Plate freezing

The plate freezer is another quick freezing technology commonly employed by seafood processors (Backi, 2015; Johnston et al., 1994). The plates are fabricated from aluminum extrusions which allow refrigerant between −30 to −50 °C to be pumped through the interior while the product is held between two refrigerated plates for a sufficient duration to enable freezing (Backi, 2015; Fellows, 2016; Hedges, 2002; Johnston et al., 1994; Kolbe & Kramer, 2007; Torry Research Station, 1977). Plate freezers employ conduction, so cooling times are typically much quicker than air blast freezers (Backi, 2015; Dempsey & Bansal, 2012; Vaclavik & Christian, 2014). Plate freezers employ a higher typical heat transfer coefficient U = 56.8 to 600 W m−2 K−1 and faster relative freezing rates up to 30 mm h−1 (Fellows, 2016; Johnston et al., 1994; Kolbe & Kramer, 2007).

Plate freezers are predominantly constructed in two formats, vertical (Fig. 3G) and horizontal (Fig. 3D), based on the orientation of the contact plates (Backi, 2015; Johnston et al., 1994; Seremeti, 2007; Vaclavik & Christian, 2014). Horizontal plate freezers work best for freezing fillets, surimi, mince, and packaged products with large flat surfaces, while vertical plate freezers are typically used for freezing whole fish into blocks both at sea and onshore (Backi, 2015; Kolbe & Kramer, 2007; Torry Research Station, 1977). The merits of plate freezers include faster cooling, a relatively small footprint, low operating costs, and minimal product dehydration (Fellows, 2016; Johnston et al., 1994). The primary detriment against plate freezers is their lack of versatility and the need for deformable and uniformly shaped products (Seremeti, 2007). Other detractions include batch operation, ice buildup potential on the plates, and products adhering to the plates during freezing (Kolbe & Kramer, 2007; Seremeti, 2007; Torry Research Station, 1977).

Immersion freezing

Immersion freezers (Fig. 3F) are an example of a direct contact freezer. They are classed as rapid freezers with freezing rates of 50 to 100 mm h−1 and higher typical heat transfer coefficient U = 210 to 290 W m−2 K−1; 680 to 740 W m−2 K−1 using forced convection (Fellows, 2016; Seremeti, 2007; Singh & Heldman, 2009). Immersion freezers operate by chilling a brine solution between −20 to −55 °C and immersing the product into it (Fellows, 2016; Johnston et al., 1994; Singh & Heldman, 2009). Paddles or a conveyor move the product from the entry to the exit end where the product is packaged. Seafood, e.g., tuna, shrimp, lobster and crab, that require rapid cooling are immersed directly in a salt brine (Kolbe & Kramer, 2007; Seremeti, 2007). CODEX standards recommend that salt pick-up, or penetration into the flesh, be kept to a minimum to avoid negatively affecting the product’s taste (FAO & WHO, 1980; Seremeti, 2007). Almy & Field (1921) reported that an average salt penetration of 2.88% on a dry weight basis was acceptable, consistent with the 3 to 4% reported by Graiver et al. (2009) for pork. Other products are packaged first before being immersed in refrigerant (Fellows, 2016). The merits of immersion freezing include high-speed freezing, efficiency and, low capital and operating costs (Fellows, 2016; Kolbe & Kramer, 2007; Singh & Heldman, 2009). The demerits include the requirement of larger pumps to counter the viscosity of low temperature brines, the potential for corrosion, the possibility of introducing off-flavours in the unpackaged product, and the toxicity of some brines (Backi, 2015; Johnston et al., 1994; Kolbe & Kramer, 2007; Seremeti, 2007).

Cryogenic freezing

Cryogenic freezers (Fig. 3E) are a type of ultrarapid freezers. They typically refer to any system where the refrigerant is sprayed directly into the freezing cabinet, comes directly in contact with the product, vaporizes and is discharged into the atmosphere (Fellows, 2016; Jessen, Nielsen & Larsen, 2014; Kolbe & Kramer, 2007). Their refrigerants are typically liquid carbon dioxide with a boiling point of −78.5 °C or liquid nitrogen with a boiling point of −196 °C with average temperatures within the cabinet between −50 to −100 °C (Backi, 2015; Boonsumrej et al., 2007; Jessen, Nielsen & Larsen, 2014). The merits of cryogenic freezers include their relatively small size, continuous operation, ability to accept nonuniform product size or shape, low capital cost, very high heat transfer coefficients approaching U = 1500 W m−2 K−1, and ultrarapid freezing rates of 100 to 1,000 mm h−1 (Fellows, 2016; Jessen, Nielsen & Larsen, 2014; Johnston et al., 1994; Kolbe & Kramer, 2007). Espinoza Rodezno et al. (2013) and Boonsumrej et al. (2007) reported lower lipid oxidation and drip loss of cryogenically frozen products after thawing. The demerits of cryogenic freezers include their high operating cost, scarcity of liquified gasses, risk of freeze-cracking, and greenhouse effects if CO2 is used as the refrigerant (Backi, 2015; Kolbe & Kramer, 2007; Seremeti, 2007).

A relative comparison of these freezing methods is summarized in Table 3.

Table 3 Overview of commercial freezing techniques.

Evaluation criteria	Still freezing	Blast freezing	Plate freezing	Immersion freezing	Cryogenic freezing	
Description	Employs cold, still air to freeze the product.	Employs cold, forced, convective air to freeze the product.	Employs compression between two chilled plates to freeze the product.	Employs immersing the product in a refrigerated bath to freeze the product.	Employs spraying liquified gas directly into the freezing cabinet to freeze the product.	
Typical temperature	−5 to −30 °C	−18 to −30 °C	−30 to −50 °C	−20 to −55 °C	−50 °C	
Typical fluid	Natural convection air	Forced convection air	Refrigerant	Brine	Liquified CO2 or N2	
Relative heat transfer coefficient	U = 5.7 W m−2 K−1	U = 22.7 to 56.8 W m−2 K−1	U = 56.8 to 600 W m−2 K−1	U = 210 to 290 W m−2 K−1 without convection and U = 680 to 740 W m−2 K−1 with forced convection	U = 1, 500 W m−2 K−1	
Merits	- Cold storage and transportation of frozen products
- Does not require uniform product size and shape	- Highspeed freezer
- Does not require uniform product size and shape
- Used in batch, semi-batch and continuous processes
- Can produce IQF product
- Air-impingement freezers have similar capabilities to cryogenic freezers at a lower cost.	- High cooling rates
- Small footprint
- Low operating costs
- Little product dehydration	- Very high cooling rates
- Does not require uniform product size and shape
- Efficiency
- Low capital and operating costs
- Continuous operation	- Ultrahigh cooling rates - Does not require uniform product size and shape
- Small footprint
- Lower capital cost
- Continuous operation
- Product is less susceptible to drip loss and lipid oxidation	
Detractions	- Slow cooling rates
- Larger, extracellular ice crystals
- Increased cellular damage and drip loss
- Not employed for commercial freezing applications
- Used for batch and semi-batch processes	- Potential for freezer burn
- Need for regular defrosting
- Cost of operating large fans	- Lack of versatility
- Needs uniform size and shape
- Ice buildup on plates
- Product adhering
- Difficult to fully automate	- Possible off-flavours
- Corrosion
- Viscosity of cold refrigerant
- Potential toxicity of some refrigerants	- Operational cost
- Availability of liquified gasses
- Risk of Freeze-cracking
- CO2 is a greenhouse gas	
Cost	Lowest cost freezer	More expensive to purchase and operate than a still freezer	Low operating costs.	Low capital and operating cost	Low capital cost. High operating cost.	
References	(Boonsumrej et al., 2007; Dempsey & Bansal, 2012; Fellows, 2016; Jessen, Nielsen & Larsen, 2013; Kolbe & Kramer, 2007; Laguerre & Flick, 2004; Norsworthy, 2015; Singh & Heldman, 2009; Suh et al., 2017)	(Alizadeh et al., 2007; Backi, 2015; Backi, 2018; Boonsumrej et al., 2007; Cheng et al., 2014; Dempsey & Bansal, 2012; Fellows, 2016; Garthwaite, 1997; Jessen, Nielsen & Larsen, 2013; Johnston et al., 1994; Kolbe & Kramer, 2007; Seremeti, 2007; Suh et al., 2017)	(Backi, 2015; Dempsey & Bansal, 2012; Fellows, 2016; Hedges, 2002; Johnston et al., 1994; Kolbe & Kramer, 2007; Seremeti, 2007; Torry Research Station 1977; Vaclavik & Christian, 2014)	(Backi, 2015; FAO & WHO 1980; Fellows, 2016; Johnston et al., 1994; Kolbe & Kramer, 2007; Seremeti, 2007; Singh & Heldman, 2009)	(Backi, 2015; Boonsumrej et al., 2007; Espinoza Rodezno et al., 2013; Fellows, 2016; Jessen, Nielsen & Larsen, 2013; Johnston et al., 1994; Kolbe & Kramer, 2007; Seremeti, 2007)	

Thawing and thawing techniques

Improved thawing methods are essential for fish processing due to the growing use of freezing as a preservation method by processors (Indzere et al., 2020; Jason, 1974). However, Haugland (2002) reported that while 75% of Norwegian processors used thawing in their production, 93% of these processors relied upon uncontrolled thawing methods. This uncontrolled thawing can provide a favourable temperature for microbes to return into active form and spoil the product (Indzere et al., 2020). Primarily, these processors batch thawed fish using either running water or air. Furthermore, there is less published research regarding commercial thawing than freezing; a Web of Science database search (https://apps.webofknowledge.com) yielded 1355 articles from 2017-2021 for “fish” AND “freezing” but only 374 articles for “fish” AND “thawing” over the same timeframe (Fig. 4).

Figure 4 Number of research articles found in the Web of Science database when searching “fish” and “freezing” vs “fish” and “thawing” published between 2017 and 2021 (conducted on April 2, 2021).

Thawing is the opposite process of freezing; the addition of heat to raise a product’s temperature to enable the subsequent phase change from a solid to liquid (Haugland, 2002; Reid, 1993). In this case, the latent heat of fusion is the amount of energy added to undergo the phase change (Vaclavik & Christian, 2014). Therefore, Eq. (1) is equally valid for thawing; however, the heat capacity is 4.18 kJ kg−1 K−1 for liquid water and 2.04 kJ kg−1 K−1 for ice. Backi (2018) also observed that freezing is faster than thawing because ice is a better conductor of heat than liquid water. Reid (1993) reported that it takes three to four times longer to thaw a product from −80 °C to +80 °C than the inverse, and Fellows (2016) further noted that the insulating effect of the water increased with thawing, further slowing the process down. Jason (1974) reported an upper temperature bound to thawing, notably that thawing above 20 °C resulted in quality challenges and thawing above 30 °C resulted in cooking; therefore, thawing must be controlled to minimize any unwanted effects such as dehydration or overheating of the flesh.

Haugland (2002) reported that the principles of heat transfer could also be applied to thawing. Heat transfer through the surface by convection or conduction and heat generation within the product using microwaves, ultrasound, dielectric methods, and electric resistance. Conventional thawing techniques employed by seafood processors include still air thawing, forced convection thawing, water sprinkling and water immersion thawing (Ragnarsson & Viðarsson, 2017). These methods use heat transfer through the surface and are inefficient, slow and inconvenient (Cai et al., 2019). Unconventional thawing methods include vacuum, humidified forced convection, high-pressure, microwave, ohmic, high voltage electrostatic field, ultrasound, and radiofrequency thawing (Cai et al., 2019; Cai et al., 2020; Coolnova, 2018; Li & Sun, 2002; Min et al., 2016; Mousakhani-Ganjeh, Hamdami & Soltanizadeh, 2015; Ragnarsson & Viðarsson, 2017; Roiha et al., 2018b; Wu et al., 2017). Regardless of the thawing process selected, processors should be wary of: bacterial growth, excess drip loss, dehydration, and localized overheating of the product (Garthwaite, 1999). Once thawed, fish are kept on ice or held in refrigerated storage until they are sold or undergo secondary or value-added processing.

Thawing in still air

Thawing in still air (Fig. 5A) employs natural convection and conduction to thaw (Backi, 2015; Haugland, 2002; Jason, 1974) and is the simplest and cheapest method commonly used by fish processors (Garthwaite, 1999; Klinkhardt, 2013; Ragnarsson & Viðarsson, 2017). Air thawing involves either storing the frozen product in ambient temperatures overnight or in chilled temperatures (<4 °C) for a prolonged period to facilitate slow thawing (Archer, Edmonds & George, 2008). Jason (1974) reported that thawing in still air is a slow process capable of thawing rates approaching 10 mm h−1 at 15 °C for whole cod. Thawing can be sped up by placing the product on grated racks to improve air circulation around the product (Archer, Edmonds & George, 2008).

Figure 5 Examples of thawing systems: (A) still air, (B) air blast, (C) warm humidified air, (D) still water, (E) automated, (F) aerated water, (G) high-pressure.

Photo and drawing credits: Pete Brown.

Thawing in still air is typically executed as a batch process that requires a significant amount of time, labour and space (Garthwaite, 1999; Ragnarsson & Viðarsson, 2017). When air thawing, temperatures should be kept below 20 °C to minimize bacterial growth; however, temperatures should not be kept too low as extended thawing periods can also lead to bacterial growth, lipid oxidation, reduced yield, and spoilage (Backi, 2015; Garthwaite, 1999; Gonçalves, Nielsen & Jessen, 2012; Regenstein & Regenstein, 1991). Lastly, when thawing in still air, a boundary layer of cold air forms around the fish, slowing thawing down and reducing control of the process (Haugland, 2002; Ragnarsson & Viðarsson, 2017).

Air blast thawing

Forced convection will help prevent the boundary layer from forming and speed up thawing (Backi, 2015; Haugland, 2002; Jason, 1974; Regenstein & Regenstein, 1991). Garthwaite (1999) recommends air speeds of 8–10 m s−1 for optimal heat transfer. Supplementing forced convection with heat will improve thawing rates further; Fig. 5B depicts the product on pallets with warm air blown over the product by circulation fans. This method is still relatively inexpensive and can improve thaw rates.

Some detractions to air blast thawing include additional capital and operating costs of the fans, increased maintenance costs and potential downtime, and the potential for hotspots and uneven air distribution (Archer, Edmonds & George, 2008). Like thawing in still air, batch processing, significant space requirements, the potential for lipid oxidation, product drying near the surface, and growth of microorganisms at higher temperatures still exist (Backi, 2015; Regenstein & Regenstein, 1991).

Thawing in humidified air

Some of the challenges associated with thawing in air, specifically lipid oxidation and dehydration, can be mitigated by humidifying the air (Backi, 2015; Jason, 1974). Furthermore, Jason (1974) reported that thawing in 20 °C humidified air moving at 8 m s−1 can produce thawing rates approaching 25 mm h−1. Indzere et al. (2020) stated that thawing in forced, humidified air was the most efficient thawing method. The Coolnova® chamber (Fig. 5C), which has found some commercial success in Europe (Coolnova, 2018), is an example of a humidified air thawing system. Product thaws rapidly in a humidified air system at a temperature of 20 to 30 °C and relative humidity of 85 to 100% due to the humid, convective, warm air. Vendors claim this method maintains the fresh quality, taste, smell and appearance of foods with minimal drip loss (Coolnova, 2018). Furthermore, these systems are available as batch systems for smaller users; and continuous systems for larger processors.

Detractions of thawing in humidified air include higher start-up and operating costs, the potential for overheating the product’s surface, the need for regular maintenance and cleaning, inconsistent programming times required due to product size and shape, and higher energy costs (Archer, Edmonds & George, 2008; Ragnarsson & Viðarsson, 2017).

Immersion in water thawing

Another widely used thawing method is water immersion (Backi, 2015; Jason, 1974; Regenstein & Regenstein, 1991). Immersion thawing can be a simple batch process (Fig. 5D); placing the product in 300–1,000 litre totes of running water and left to thaw to a computer-controlled, self-unloading, automated process (Fig. 5E). Backi (2018) reported thawing rates of 25–34 mm h−1 at a constant temperature of 15.5 °C. Immersion thawing should incorporate a recirculation system, such as aeration, to conserve heat and water (Fig. 5F). To thaw the product in aerated water, place the frozen fillets in a tub of water and bubble air into the tub to circulate the water using a manifold placed in the tank connected to an air compressor. Lipid oxidation and surface drying were minimized or eliminated using this thawing technique (Backi, 2018).

The potential challenges with thawing in water include absorption of water and subsequent loss of flavour (Gonçalves, Nielsen & Jessen, 2012; Jason, 1974), product contamination (Jason, 1974; Regenstein & Regenstein, 1991), potential yield loss in partially thawed product, and water usage, treatment, and disposal costs (Archer, Edmonds & George, 2008).

Ultrahigh-pressure thawing

There has been significant research on applying ultrahigh-pressure thawing techniques to foodstuffs (Cui et al., 2019; De Alba et al., 2019; Rouillé et al., 2002; Rubio-Celorio et al., 2015; Wen et al., 2015; Wu et al., 2017). The effects of ultrahigh-pressure thawing are typically associated with temperature, pressure, and pressure change rate (Backi, 2015; Cai et al., 2019; Chevalier et al., 1999; Wu et al., 2017). At higher pressure (Fig. 6), the freezing point of liquid water drops from 0 °C at atmospheric pressure to −21 °C at 210 MPa (Backi, 2018; Wu et al., 2017). Chevalier et al. (1999) reported that the effective thawing time for fillets in a 10 °C water bath decreased four-fold between atmospheric pressure and 200 MPa; Rouillé et al. (2002) obtained optimal thawing results at a pressure of 150 MPa. Ultrahigh-pressure systems used in the food processing industry (Fig. 5G) tend to be high-precision, mechanized devices that use hydraulic intensifiers to increase pressure and require significant maintenance (Archer, Edmonds & George, 2008; Backi, 2018). The merits of ultrahigh-pressure thawing include improved water holding capacities and less drip loss during the thawing process (Ragnarsson & Viðarsson, 2017; Rouillé et al., 2002; Wu et al., 2017).

Figure 6 Temperature-pressure phase diagram for water.

Ultrahigh-pressure thawing is a novel technology with limited practicality due to its high costs (Indzere et al., 2020). Archer, Edmonds & George (2008) also reported that tuna fillets thawed using ultrahigh-pressure resulted in unfavourable colour changes.

A relative comparison of the thawing methods presented above is summarized in Table 4.

Table 4 Overview of commercial thawing techniques.

Evaluation Criteria	Still air thawing	Air blast thawing	Thawing in humidified air	Immersion thawing	Ultrahigh pressure thawing	
Description	Employs natural convection and conduction to thaw.	Employs forced convection and conduction to improve thawing rates.	Employs forced convection of heated and humidified air to improve thawing rates.	Employs a bath of running water to thaw product more quickly. Aerating the water can improve thawing.	Employs a water bath under ultrahigh pressure (∼210 MPa) and allowed to thaw more rapidly and at lower temperatures.	
Thawing rate	10 mm h−1 at 15 ° C		25 mm h−1 at 20 ° C and 8 m s−1	25-34 mm h−1 at 15.5 ° C	<100 mm h−1 at 10 ° C and 200 MPa	
Merits	- Is simple to set up.
- No expensive equipment to purchase.
- Using racks improved air circulation and thawing rate.	- Prevents boundary layer from forming.
- Improved thawing rates.	- Efficient thawing.
- Fresh quality, taste, smell, and appearance of foods are protected.
- Less drip-loss after freezing and thawing.
- Potential for automation.	- Faster than thawing in air.
- Less yield loss.
- Minimizes lipid oxidation.	- High heat transfer rates.
- Better water holding capacity.
- Less drip loss	
Detractions	- Slow process.
- Requires ample space.
- Labour-intensive batch process.
- Product may suffer from surface dehydration, lipid oxidation and growth of microorganisms.	- Requires ample space.
- Labour-intensive batch process.
- Additional capital and operating costs of the fans.
- Increased maintenance costs.
- Potential downtime.
- Potential for hotspots.	- Higher capital and operating costs.
- Higher energy costs.
- Maintenance and cleaning costs.
- Potential for overheating the product surface.
- Inconsistent programming times.
- required and	- Absorption of water and loss of flavour.
- Water use and treatment costs.
- Potential contamination.
- Potential yield loss.	- Very high capital and operating costs.
- Significant maintenance and potential downtime.
- Possible colour change.	
Cost	Lowest cost.	More expensive than still-air thawing.	Higher capital, maintenance and operating costs compared to air blast thawing.	Higher cost compared to thawing in air.	Highest cost solution.	
References	(Archer, Edmonds & George, 2008; Backi, 2015; Garthwaite, 1999; Haugland, 2002; Jason, 1974; Klinkhardt, 2013; Ragnarsson & Viðarsson 2017)	(Archer, Edmonds & George, 2008; Backi, 2015; Garthwaite, 1999; Haugland, 2002; Jason, 1974; Regenstein & Regenstein, 1991)	(Archer, Edmonds & George, 2008; Backi, 2015; Coolnova, 2018; Indzere et al., 2020; Jason, 1974; Ragnarsson & Viðarsson 2017)	(Archer, Edmonds & George, 2008; Backi, 2015; Backi, 2018; Gonçalves, Nielsen & Jessen, 2012; Jason, 1974; Regenstein & Regenstein, 1991)	(Archer, Edmonds & George, 2008; Backi, 2015; Backi, 2018; Cai et al., 2019; Chevalier et al., 1999; Indzere et al., 2020; Ragnarsson & Viðarsson 2017; Rouillé et al., 2002; Wu et al., 2017)	

To choose the correct preservation treatment, processors need to understand the market and consumers’ expectations. For example, freezing is the primary preservation method Newfoundland and Labrador’s cod processors use to bridge the gap between the timing and location of their fisheries and the clients’ needs for a thawed product that compares closely to fresh. However, freezing techniques have their limitations as degradation of organoleptic and sensory qualities such as taste, texture, and juiciness if the freezing and thawing methods are not optimized. The correct application of freezing throughout the cold chain combined with an optimized thawing process is needed to maximize quality and increase value.

Effect of freezing and thawing on the quality of seafood

Freezing is an efficient method to maintain seafood quality and maintain the product’s fresh taste (Dang et al., 2018a). However, as already reported, some deterioration can still occur during freezing. Therefore, seafood quality and storage life depend on several variables, including species, initial product quality, initial microbial load, handling, freezing rate, storage temperature, packaging, thawing method and cold chain abuse (Baygar & Alparslan, 2015; Dang et al., 2018a; Dawson, Al-Jeddawi & Remington, 2018; Fuentes et al., 2013). Improper freezing and thawing techniques can result in quality defects, including poor colour and texture and freezer burn (Chevalier et al., 2000; Frelka et al., 2019). Slow freezing of cod results in larger ice crystal size, creating network structures in the flesh and resulting in a spongy texture after thawing (Nakazawa & Okazaki, 2020). Improper long-term cold storage can result in protein denaturation, lipid oxidation and rancidity (Ghaly et al., 2010; Kolanowski, Jaworska & Weißbrodt, 2007; Lorentzen et al., 2019; Nakazawa & Okazaki, 2020). Cold-chain abuse and sublimation can result in the thawing and recrystallization of ice, resulting in larger, more irregular crystals. These ice crystals can increase damage to the muscle structure and reduce product quality by decreasing water holding capacity and juiciness (Dawson, Al-Jeddawi & Remington, 2018; Tironi, de Lamballerie & Le-Bail, 2010); the loss of bound water is referred to as drip loss.

Global value chains (GVC) within the seafood industry have been discussed in detail (Jensen & Sørensen, 2016; Knútsson, Kristófersson & Gestsson, 2016; Norsworthy, 2015; Phyne, Hovgaard & Hansen, 2006; PrimeFish, 2017; Trondsen, 2012; Witter & Stoll, 2017). The GVC for seafood is defined as the activities needed to bring a product from capture or harvest to the final consumer. These activities may vary based on species, finished product, region, and vertical integration within the sector (Kaplinsky, 2004; PrimeFish, 2017). Witter & Stoll (2017) presented a typical GVC for seafood (Fig. 7), which identifies potential GVC activities, and each of these interactions can potentially impact the quality of the final product (Secci & Parisi, 2016). The primary steps identified in typical GVC cold chains include post-processing, distribution, marketing, secondary processing, and final distribution; not all activities in the GVC apply to all fisheries.

Figure 7 Generalized seafood value chain depicting frozen, processed seafood being sold in a different state, province or country than where it was landed.

Image data sources: Knútsson, Kristófersson & Gestsson, 2016; Witter & Stoll, 2017.

Outsourcing significant value-added processing to lower-cost countries such as China, Vietnam, and Lithuania helps minimize costs (PrimeFish, 2017; Stringer, Simmons & Rees, 2011; Verge, 2017) and is partially responsible for additional freezing and thawing. Therefore, for this paper, North Atlantic commercial fisheries will refer to wild-caught or farmed species prosecuted in North Atlantic waters by European, Canadian and United States interests with a minimum acceptable amount of domestic or onboard processing. The main commercial sectors for the North Atlantic fisheries include demersal, pelagic, and shellfish species (DFO, 2018a; National Marine Fisheries Service, 2017; PrimeFish, 2017). This study will disregard other sectors such as roe, marine mammals, and marine plants due to their relative sizes or the lack of available information.

Demersal species

Demersal species refer to fish that dwell or feed on or near the seabed (Bender, 2014a). These species, commonly referred to as groundfish or whitefish, fall into two categories: round (i.e., cod or halibut) or flat (i.e., flounder or turbot). Bayliss (1996) reported that the flesh of demersal fish is translucent grey-white due to low myoglobin content and that it only contains up to 2% fat because demersal fish store the oil in their livers. Several key, commercial, North Atlantic groundfish species include Atlantic cod (Gadus morhua), Ocean perch (Sebastes marinus), Golden redfish (Sebastes norvegicus), Silver hake (Merluccius bilinearis), Atlantic halibut (Hippoglossus hippoglossus), Haddock (Melanogrammus aeglefinus), Summer flounder (Paralichthys dentatus), and Saithe (Pollachius virens) (DFO, 2018a; ICES, 2019; National Marine Fisheries Service, 2017; Norsworthy, 2015). Commercially, groundfish is caught using longlines, trawls, pots/traps, seines and gill nets (Jennings, Kaiser & Reynolds, 2001; Montgomerie, 2015). Seasonality and gear choice can affect quality; however, these topics fall outside the scope of this review paper.

Key groundfish species caught in North Atlantic fisheries include cod, halibut, haddock, flounder, and turbot. Each of these species supports commercial fishing in Europe and North America (DFO, 2018a; National Marine Fisheries Service, 2017; PrimeFish, 2017), with a mixture of fresh and frozen landings (ICES, 2019; National Marine Fisheries Service, 2017; PrimeFish, 2017). The GVC for groundfish fisheries can vary by species, region, season, capture method and degree of processing; however, they typically adhere to the model presented in Fig. 7 (Knútsson, Kristófersson & Gestsson, 2016; Witter & Stoll, 2017).

After harvest, removing the gut material and cleaning the internal cavity is typically carried out to minimize autolytic and microbiological degradation (Ghaly et al., 2010; Hanenian & Mittal, 2004). A shelf life of up to 12 days can be achieved for groundfish products by maintaining chilled temperatures (1 to 5 °C) and employing good handling processes from catch to market (Castell, 1954; Olafsdóttir et al., 2006). The shelf-life can be extended to 15 days by super-chilling to −1.5 °C immediately after harvest (Bayliss, 1996; Dang et al., 2018a; Duun & Rustad, 2007; Olafsdóttir et al., 2006). Therefore, if landings cannot be processed, shipped, purchased, and used by the end client within two weeks post catch, freezing, holding, and thawing the product can extend the freshness by further slowing down microbial activity. To provide more detail into the GVC for demersal species, fresh and frozen groundfish landings are summarized in the following subsections.

Fresh groundfish landings

Post catch, the fish are bled and gutted onboard the fishing vessel to remove the natural enzymes and help mitigate against autolytic degradation (Huss et al., 1995). The fish are then stored in the vessel’s hold or in pans using ice slurry or flake ice to keep it chilled and maintain freshness until the processor receives them; thus, minimizing microbial degradation (Gram & Dalgaard, 2002). Once landed, primary processing (heading, filleting, and skinning) occurs, and fillets are sold into the fresh market. Secondary, value-added processing such as portioning, mincing, breading, and consumer packaging may also follow at the same processor or in a different jurisdiction by a lower-cost processor before being sold into the fresh market (Stringer, Simmons & Rees, 2011).

The final product may also be frozen to slow down autolytic and microbiological degradation and extend freshness (Benjakul & Bauer, 2000; Ghaly et al., 2010; Haugland, 2002). Proper freezing and cold storage will allow the product to be held and sold when demand and price are higher or shipped long distances using low-cost ground transport to larger markets (Ziegler et al., 2013). Frozen fillets can also be sent to a secondary processor to complete the value-added work, such as portioning, coating, or preparing ready-to-serve products. After processing, the fillets can be sold as frozen or chilled refreshed (Mai, Nguyen & Nguyen, 2020). Today, much value-added work is completed in developing countries to reduce costs, and final products can be refrozen and shipped back (Stringer, Simmons & Rees, 2011).

As shown in Fig. 7, commercially harvested groundfish can be frozen multiple times along the value chain. This process is termed “twice-frozen” or “double frozen” (Hedges, 2002). This repeated freezing and thawing can affect product quality if proper care is not taken (Fuentes et al., 2013; PrimeFish, 2017; Washburn et al., 2017) and may result in dehydration, drip loss, and gaps in the flesh (Dawson, Al-Jeddawi & Remington, 2018; Lorentzen et al., 2019; Tironi, de Lamballerie & Le-Bail, 2010).

Frozen groundfish landings

Groundfish is landed frozen primarily in offshore fisheries and regions with large commercial fisheries and little vertical integration (PrimeFish, 2017). Freezer trawlers land frozen products with some degree of processing completed onboard. Frozen fillets or headed and gutted (H&G) products are produced quickly at sea post-catch to preserve the initial quality (Hedges, 2002; Norsworthy, 2015).

Onboard processing has the distinct advantage of allowing the fish to go through Rigour-Mortis (rigour) after processing occurs and the animals are frozen. Rigour is the binding of the myofibrils’ thick and thin filaments (Hedges, 2002). This binding occurs through the coupling of actin and myosin by consuming adenosine triphosphate as energy and results in the loss of muscle extension (Yao et al., 2019). Peters et al. (1968) found that freezing fish pre-rigour produced a higher quality product than freezing in- or post-rigour. However, Hedges (2002) noted that if thawing occurs before rigour is complete, the fish may complete rigour during the thawing process, which will result in excessive drip loss and damage to the flesh.

After landing, the frozen product is shipped for secondary processing to low-cost countries. The final products are refrozen and shipped to the end consumers (PrimeFish, 2017; Stringer, Simmons & Rees, 2011). Similar to fresh groundfish landings, frozen landings can be frozen and thawed several times along the GVC. Therefore, proper freezing of groundfish along the GVC can reduce cost and improve quality and yield.

Pelagic species

Pelagic species refer to fish that dwell and feed near the surface or the mid-water column (Bender, 2014b). Bayliss (1996) reported that because pelagic fish are more active than groundfish, their flesh has a more aerobic metabolizing reddish hue, containing up to 35% lipids and makes them a rich source of polyunsaturated fatty acids. Large, commercial, North Atlantic pelagic fisheries include Bluefin tuna (Thunnus thynnus), Atlantic herring (Clupea harengus), Atlantic mackerel (Scomber scombrus), Atlantic salmon (Salmo salar), European anchovy (Engraulis encrasicolus), and Capelin (Mallotus villosus) (DFO, 1980; DFO, 2019a; ICES, 2019; National Marine Fisheries Service, 2017; Norsworthy, 2015). Pelagic fish are caught using seines, trawls, drift nets, fish traps, and long-lines (Jennings, Kaiser & Reynolds, 2001; Montgomerie, 2015). Atlantic Salmon are also farmed by commercial aquaculture operations throughout the region (National Marine Fisheries Service, 2017; PrimeFish, 2017).

Key pelagic species in North Atlantic fisheries such as tuna, herring, mackerel, and salmon are harvested from wild capture and aquaculture fisheries (DFO, 1980; DFO, 2019a; National Marine Fisheries Service, 2017; PrimeFish, 2017). Pelagic species can suffer from lipid oxidation, colour change, colour fading, formaldehyde formation, and protein denaturation if not preserved effectively (Nakazawa & Okazaki, 2020). Pelagics typically have higher lipid concentrations in their flesh than groundfish, so rancidity due to oxidation can be a primary concern during cold storage (Bayliss, 1996; Devadason et al., 2016; Hedges, 2002; Papuc et al., 2017). As a result, many pelagics are traditionally preserved by salting, drying, smoking, pickling, and canning; alternatively, salmon and herring are regularly frozen along the GVC (DFO, 1980; PrimeFish, 2017). More details into the GVC for pelagic fisheries, wild-capture and aquaculture fisheries are presented in the following subsections.

Wild capture pelagic species

Atlantic herring is one of the most abundant wild-capture pelagic fisheries by weight in the North Atlantic and possesses a relatively simple GVC (DFO, 2018a; Gudmundsson, Asche & Nielsen, 2006; National Marine Fisheries Service, 2017; PrimeFish, 2017). Bayliss (1996) reported that pelagic fish such as mackerel and herring only have a seven-day shelf life at refrigerated temperatures, and as a result, preservation is necessary to maintain freshness.

The primary markets for herring are Japan, Western Europe, and Africa (Gudmundsson, Asche & Nielsen, 2006). Over the past decade, consolidation within the European herring fishery has been due to the reduced prices paid for herring and reduced quotas (PrimeFish, 2017). In North America, catches also declined, but these catches’ value increased (National Marine Fisheries Service, 2017). As a result, herring’s Icelandic and Norwegian production is reduced to frozen fillets and whole, frozen, H&G fish exported for sale or secondary processing elsewhere (PrimeFish, 2017).

Processing of frozen Atlantic herring is presented in detail by Dang et al. (2018b), Hamre, Lie & Sandnes (2003), and Tolstorebrov, Eikevik & Indergård (2014). Freezing herring extends the product’s shelf life by slowing down autolytic and microbiological degradation (Benjakul & Bauer, 2000; Ghaly et al., 2010; Haugland, 2002). Holding at very low temperatures minimizes colour change and fading (Bito, 1976; Nakazawa & Okazaki, 2020). Care should be taken throughout the freezing and thawing process to maximize water holding capacity and minimize freezer burn, texture loss, and gaping (Dawson, Al-Jeddawi & Remington, 2018; Lorentzen et al., 2019). Like groundfish, rapid freezing minimizes ice crystal size and minimizes damage to the flesh (Dawson, Al-Jeddawi & Remington, 2018; Tironi, de Lamballerie & Le-Bail, 2010). Pre-rigour freezing reduced thawing loss and oxidation better than in- and post-rigour freezing (Dang et al., 2018b), and cold-chain abuse must be avoided to maintain shelf-life. Hamre, Lie & Sandnes (2003) noted that the dark muscle was three to four times more susceptible to oxidation than the light muscle and should be removed before freezing to improve quality. Tolstorebrov, Eikevik & Indergård (2014) also found that cold-storing herring at −45 °C inhibited oxidation compared to −25 °C; he also found that vacuum packaging herring fillets in a medium oxygen barrier coupled with cold storage could produce similar results at −25 °C. Pelagics, such as tuna, have high concentrations of myoglobin in their flesh which gives the meat a reddish hue; to prevent the oxidation of the myoglobin and subsequent loss of colour. Previous studies have shown that these species need to be cold stored at temperatures below −35 °C (Bito, 1976; Nakazawa & Okazaki, 2020). Therefore, processing and freezing pelagic species pre-rigour, removing dark muscle during processing, cold storing at ultralow temperatures, and correct packaging selection can enhance the fillets’ storage quality and improve yield.

Pelagic aquaculture

Atlantic salmon is primarily an aquaculture fishery managed by large, vertically integrated, multinational corporations and is the largest pelagic fishery by value in the North Atlantic region (DFO, 2018a; National Marine Fisheries Service, 2017; PrimeFish, 2017). The GVC for salmon is similar to groundfish; however, due to the full integration and enhanced processing technologies available, more secondary and value-added processing occurs before shipping (Miller, 2017; Norsworthy, 2015), minimizing the need for multiple freeze and thaw cycles. Most salmon produced by aquaculture companies are sent to the fresh market as whole, gutted, headed and gutted (H&G), fillets, portions and minced products. Some material is frozen and shipped to value-added processors elsewhere, e.g., frozen salmon portions produced in the Faroe Islands for European and US markets (PrimeFish, 2017).

Alizadeh et al. (2007), Dawson, Al-Jeddawi & Remington (2018), and Sigurgisladottir et al. (2000) discussed processing frozen Atlantic salmon in detail. The colour change and weight loss found in frozen and thawed salmon were downsides of freezing and occurred because of damage to the cellular structure (Alizadeh et al., 2007; Dawson, Al-Jeddawi & Remington, 2018). Faster freezing rates reduced this damage. Dawson, Al-Jeddawi & Remington (2018) also reported that oxidation increases with cold storage time. Alizadeh et al. (2007); Dawson, Al-Jeddawi & Remington (2018) reported freezing as a vital preservation method to extend the shelf life of salmon. The authors also noted that increasing the freezing rate increased the number of nucleation points, reduced the ice crystals’ size, and produced a higher quality frozen product with less drip loss, better texture, and protein stability. Unlike many other pelagic species, Atlantic salmon demonstrated slight quality deterioration when stored at −25 °C vs. −40 °C, and special packaging has been shown to improve product quality (Indergård et al., 2014). Alizadeh et al. (2007) found that more, smaller ice crystals were produced when high-pressure processing (HPP) was used in conjunction with freezing and thawing resulting in even less drip loss. Therefore, the quality and yield of frozen and thawed salmon can be improved by increasing the freezing rate in conjunction with using HPP. Minimizing the time expended in cold storage reduces colour change and rancidity due to oxidation.

Shellfish species

Shellfish refers to a wide range of marine mollusks, echinoderms and crustaceans (Bender, 2014c). Lipid content is 0.8–2.3% in mollusks and 0.8–6.7% in crustaceans such as shrimp, lobster and crab (Newcombe, 1944; Sirot et al., 2008). Large, commercial, North Atlantic shellfish fisheries include American lobster (Homarus americanus), Snow crab (Chionoecetes opilio), Dungeness crab (Metacarcinus magister), Orange-footed sea cucumber (Cucumaria frondosa), Atlantic sea scallops (Placopecten magellanicus), Atlantic surf clams (Spisula solidissima), Hard clams (Mercenaria mercenaria), Blue mussels (Mytilus edulis), Atlantic white shrimp (Litopenaeus setiferus), and Northern shrimp (Pandalus borealis) (DFO, 2018b; DFO, 2019b; National Marine Fisheries Service, 2017; PrimeFish, 2017). Crustaceans such as lobster and crab are typically caught using baited traps or pots; shrimp are caught using beam-trawls (Montgomerie, 2015). Marine mollusks such as clams, oysters, mussels and scallops, and echinoderms such as sea cucumber are typically harvested using a dredge (Montgomerie, 2015). The annual global production of marine bivalves for human consumption is 15 million tonnes, with 89% farmed in commercial aquaculture operations (Wijsman et al., 2019).

Key shellfish species in the North Atlantic fishery include lobster, crab, shrimp, sea cucumbers, scallops, clams, and mussels. Shellfish are reported to have a wide range of lipid concentrations depending on species (Newcombe, 1944; Sirot et al., 2008; Zhong, Khan & Shahidi, 2007), so oxidation is still a challenge and must be considered when held in cold storage.

Shellfish are typically landed live, processed, and shipped to market. Crustaceans and mollusks can be processed and sold live, cooked, or frozen (National Marine Fisheries Service, 2017; Norsworthy, 2015; Sherstneva, 2013); echinoderms are typically sold cooked and frozen, smoked, or dried (Zhong, Khan & Shahidi, 2007). Frozen shellfish can be retailed or shipped for secondary processors elsewhere (Stringer, Simmons & Rees, 2011). The impact of freezing and thawing on the GVC for shellfish, mollusks, echinoderms, and crustaceans is presented in the following subsections.

Mollusks

Mollusks such as scallops, mussels and clams are landed live and processed. Khan & Liu (2019) reported that freezing has a limited impact on mollusks’ shelf life. Therefore, processors must combine freezing with other thermal treatments, modified atmospheric packaging (MAP) and HPP for long-term storage. This process is necessary to improve microbial safety, minimize spoilage, and maintain the product’s fresh flavour (Narwankar et al., 2011). Like finfish, care must be taken during freezing and thawing to mitigate the damage caused by cold chain abuse and reduce hold times in cold storage to reduce oxidation (Baker, 2016).

Echinoderms

Echinoderms, such as sea cucumbers, are considered an essential food in the Indo-Pacific region because of their antioxidant and medicinal properties (Zhong, Khan & Shahidi, 2007). Sea cucumber, for example, is landed live, gutted to minimize autolytic and microbiological spoilage, and processed for sale. Processing consists of cooking, drying, smoking, packaging and freezing (Zhong, Khan & Shahidi, 2007). Frozen product is shipped to Southeast Asia markets where it is either sold frozen or thawed for secondary processing. Again, care must be taken during freezing and thawing to minimize cold chain abuse.

Crustaceans

Crustaceans such as lobster, crab and shrimp are high-value species in many North Atlantic fisheries (National Marine Fisheries Service, 2017; Norsworthy, 2015; Sherstneva, 2013; Steneck et al., 2011). Shrimp are primarily landed fresh or frozen (DFO, 2018a). Once landed, they are cooked, peeled, frozen, and packaged or canned (DFO, 2018a; National Marine Fisheries Service, 2017; Norsworthy, 2015). Lobster and crab are landed live due to their short shelf-life and sold live or cooked and frozen. Crab processing includes butchering, cooking, chilling, freezing, packaging, and cold storing; frozen lobster is almost identical (National Marine Fisheries Service, 2017; Norsworthy, 2015). Freezing is a valuable preservation technique for crustaceans. Immersion freezing is the primary method to rapidly reduce the product from its cooking temperature (80 °C to 90 °C) to −18 °C quickly to maintain quality. However, like other species, freezing and frozen storage may lead to the denaturation of the myofibrillar proteins, resulting in textural changes such as reduced juiciness and water holding capacity (Lorentzen et al., 2019). Extended use of cold storage may also result in lipids’ oxidation (Papuc et al., 2017).

Summary of freezing and thawing on commercial North Atlantic seafood species

The merits and detractions of freezing and thawing of commercial North Atlantic Seafood species are summarized in Table 5.

Table 5 Summary of freezing and thawing commercial North Atlantic seafood species.

	Demersal Species	Pelagic Species	Shellfish Species	
Habitat	Near Bottom	Mid to high in the water column	Seafloor	
Activity Level	Sedentary	Highly active	Inactive	
Lipid content in the flesh	Low	High	Varies by species	
Merits of freezing	Increased shelf life allows the product to be shipped to lower-cost regions for secondary processing or larger markets, allows the product to be held until demand improves, freezing before rigour can enhance product quality.	Increased shelf life allows the product to be shipped to lower-cost regions for secondary processing or larger markets.	Allows echinoderms to be shipped to market after processing, will enable crustaceans to be cooled quickly and shipped for secondary processing or market.	
Detractions of freezing	Drip loss, protein denaturation, lipid oxidation, and texture changes if the temperature is not maintained below −30 °C.	Drip loss, lipid oxidation, denaturation of proteins, discolouration of flesh, not effective for some species (i.e., capelin, anchovies, and sardines).	Not effective for mollusks, it can result in drip loss, textural changes in crustaceans due to protein denaturation, and lipid oxidation if not adequately performed.	

Assessment of the Newfoundland and Labrador Cod Fishery

Declining shellfish stocks

The fishing industry in Newfoundland and Labrador is in transition. Fisheries and Oceans Canada (DFO) reported that Northern shrimp (Pandalus borealis), Striped shrimp (Pandalus montagui), and Snow crab (Chinoecetes opilio) stocks, which have supported the local fishing industry since the collapse of the groundfish stocks, are declining due to intense fishing pressures and climate change (DFO, 2014; DFO, 2016; DFO, 2017a; DFO, 2021b; DFO, 2021c). Recent DFO surveys estimated crab biomass (except for the North Atlantic Fisheries Organization (NAFO) Divisions 2H and 2J (Fig. 8) and reported stocks remain near historic low levels (DFO, 2021b) while multispecies trawl surveys by DFO (2017a) and DFO (2021c) indicated a significant decline in Northern shrimp and Striped shrimp fishable biomass indexes. These declines have resulted in recent cuts to crab and shrimp quotas for Newfoundland and Labrador processors and harvesters.

Figure 8 Map of Newfoundland and Labrador NAFO regions.

Image data sources: Becker et al. 2018; Bivand, Keitt & Rowlingson, 2021; Bivand & Lewin-Koh, 2021; Hijmans 2021; McIlroy et al. 2020; R Core Team 2021; Wickham, 2016; https://www.nafo.int/Data/GIS.

While cold water species stocks declined, DFO (2017b) reported that the Northern cod stock total biomass increased from 9.8 kt in 1995 to 481.3 kt in 2016 in NAFO Divisions 2J, 3K and 3L (Fig. 8). DFO (2019b) and DFO (2021a) also reported that the spawning stock biomass (SSB) increased to 398 kt in 2019; however, it still only represents 48% of the limit reference point established by DFO (2011) and in the critical range. A sizeable commercial fishery is potentially still a decade away (CCFI 2017); however, the industry needs to prepare now for its future success. Therefore, it is necessary to assess the readiness of Newfoundland and Labrador’s fisheries to re-enter the cod supply chain as a significant global supplier and generate the best value from these limited resources to help grow the provincial economy.

Challenges for rebounding cod stocks

Since the Minister of Fisheries declared a moratorium on the Northern cod fishery in NAFO Divisions 2J, 3K, and 3L on July 2, 1992 (Crosbie, 1992), there have been significant changes in the global market for whitefish. Cod block was the primary product for the Newfoundland and Labrador processors before the moratorium, and it was used to manufacture breaded, battered, and frozen products such as fish sticks and other low-end products. Since 1992, cod block has mostly been replaced by Alaskan Pollock and cod block produced as a by-product from processing cod fillets (i.e., trimming discards and poor-quality fillets) (Tonkovitch, 2017; Verge, 2017). Furthermore, the mid-grade white fish product market (quick service and casual dining establishments) is primarily being supplied using Pollock (Tonkovitch, 2017) and twice-frozen whitefish fillets processed in China, Vietnam and Lithuania (Hedges, 2002; PrimeFish, 2017; Verge, 2017). Low-cost fresh options are produced from inexpensive farmed fish such as pangasius catfish and tilapia (Verge, 2017). Therefore, local processors have limited export options and should focus on the high-end, fresh market driven by consumers looking for high-quality, fresh products with extended shelf lives (Ashie, Smith & Simpson, 1996). Iceland’s cod fishery currently dominates this market for fresh cod products; however, successfully penetrating this market will allow the Newfoundland and Labrador industry to earn the most value from its cod resources.

Therefore, Newfoundland and Labrador cod harvesters and processors need to evaluate strategies to penetrate the high-end market and maximize the value from the limited Northern cod resources by providing a timely and high-quality product to the market (Brown, 2018). However, before Newfoundland and Labrador processors can enter this market, several challenges first need to be mitigated to maximize the value of Atlantic cod, including: (a) Newfoundland’s cod fishery is dominated by small, inshore boats with limited space for ice production or ice storage (NL-GIDC, 2017); (b) the peak fishing season for these small vessels is roughly four months-long between May and September (NL-GIDC, 2017) and does not correspond with the peak consumer demand (Lent: typically, February-April), which reduces stakeholder value (Sackton, 2014); and (c) Newfoundland is an island in the North Atlantic with limited, timely options for commercial transportation of fresh product to national and international markets. Furthermore, the weather during the winter is dominated by strong winds, heavy precipitation, and intense fog. These extreme weather conditions further limit commercial transportation reliability during the winter when the demand and price for cod are highest. Therefore, until the industry can effectively deal with the space and time logistical challenges mentioned above, Newfoundland and Labrador processors can expect to supply frozen cod to the market (Brown, 2018). The supply of frozen fillets results in a fourth challenge (d) the price paid for frozen cod fillets is approximately half compared to fresh cod fillets, as shown in Fig. 9 (Sackton, 2014) due to the perceived and actual reduction in sensory quality discussed in earlier sections.

Figure 9 Price ladder for cod products in US$.

Image data source: Sackton, 2014.

Employing freezing and thawing to improve quality

When done correctly, freezing and thawing cod fillets will increase shelf life, improve food safety, reduce shipping costs, and level out the seasonal variabilities between supply and demand (Backi, 2015; Haugland, 2002; NL-GIDC, 2017; Torry Research Station 1977). However, cold-chain abuse and improper freezing and thawing techniques will result in quality defects, including protein denaturation, oxidation, poor colour and texture, and freezer burn, which will result in the deterioration of the fillet quality (Chevalier et al., 2000; Frelka et al., 2019; Leygonie, Britz & Hoffman, 2012; Nakazawa & Okazaki, 2020).

Jensen et al. (2010) reported that a chilled product (frozen and thawed) could receive higher prices than a frozen product if the quality of the chilled product closely resembles a fresh product. Therefore, value can be maximized by: (a) employing processes that improve freezing rates and minimize ice crystal size, (b) minimize protein denaturation and oxidation during cold storage, (c) minimize the number of freeze-thaw cycles, and (d) effectively manage the cold chain throughout the entire GVC. Hence, Newfoundland and Labrador processors could employ these principles: (a) to produce a high-quality frozen product, (b) hold it until demand and price improve, (c) transport product to larger markets using lower-cost ground transportation, (d) thaw the products using highly controlled processes and (e) sell them during peak season as chilled which will command a price closer to fresh.

Freezing and thawing foodstuffs can be divided into three phases: freezing, cold storage and thawing (Hanenian & Mittal, 2004). As discussed earlier, freezing occurs when a product undergoes a phase change from liquid to solid. Typical freezing methods employed by Newfoundland cod processors act on the boundary and include air blast, plate, immersion, and tunnel freezing. Spray and cryogenic freezing are also popular boundary methods (Torry Research Station 1977); however, restricted in Newfoundland due to the scarcity and cost of liquified gasses. Local seafood processors do not use instantaneous methods such as magnetic, pressure, and cell alive freezing, and these systems will not be considered further. Therefore, given these limitations, to improve the quality of frozen cod fillets, processors need to: (a) employ rapid freezing techniques to minimize ice crystal size and subsequent damage to the cellular structure; (b) store product below −30 °C to minimize oxidation and protein denaturation; and (c) minimize cold chain abuse during the holding and transport phases.

As discussed above, thawing is the addition of heat to raise the product’s temperature and enables the subsequent phase change from a solid to a liquid (Backi, 2015; Haugland, 2002; Reid, 1993). Again, thawing methods fall into two main categories: methods through the boundary and methods acting on the inner domain (Backi, 2015; Haugland, 2002). Thawing methods typically employed by local processors act on the boundary and include thawing in air and water. Other thawing methods that act on the boundary include vacuum and contact thawing; however, they are not typically used locally in Newfoundland and Labrador. Methods acting on the inner domain include ultra-high pressure, dielectric, electrical resistance and microwave (Backi, 2015). Local fish processors do not use freezing methods acting on the inner domain. Therefore, to improve the quality of thawed products: (a) thawing should not take place until the product is entirely out of rigour; (b) thawing methods should be rapid and employ circulation of the thawing medium to minimize dehydration; (c) thawing methods should not promote the growth of microorganisms; and (d) thawing methods can employ ultrahigh pressure to maximize the quality parameters.

Summary

There has been considerable research into the spoilage mechanisms and subsequent preservation of seafood (Cheng et al., 2014; Ghaly et al., 2010; Persson & Löndahl, 1993; Seremeti, 2007; Suh et al., 2017). The spoilage of seafood is broken down into three phases: autolytic degradation, bacterial spoilage, and oxidation (Ghaly et al., 2010; Huss et al., 1995; Surette, Gill & LeBlanc, 1988; Tull, 1996). Autolysis is responsible for textural changes to the flesh and degradation of freshness with little microbial change (Benjakul & Bauer, 2000; Gram & Huss, 1996; Haugland, 2002; Surette, Gill & LeBlanc, 1988). Microbial spoilage typically follows autolysis and may degrade taste, colour, odour, texture, and juiciness. Microbial degradation in seafood also results in products that are not fit for human consumption leading to unnecessary waste (Cheng et al., 2014; Ghaly et al., 2010; Gram & Dalgaard, 2002). Lastly, seafood held in cold storage for an extended period or at temperatures that are not sufficiently low can suffer from oxidation, especially if lipid content in the product is high (Ghaly et al., 2010; Kolanowski, Jaworska & Weißbrodt, 2007; Nakazawa & Okazaki, 2020; Papuc et al., 2017).

Seafood is at its peak freshness immediately after death and must be preserved quickly to maintain its freshness (Huss et al., 1974; Lakshmanan, 2000; Warrier, Gore & Kumta, 1975). Preservation entails slowing down the food spoilage process, and humans have been preserving food for centuries (Kaloyereas, 1950; Vaclavik & Christian, 2014). Conventional preservation techniques used by the seafood industry include salting, drying, smoking, pickling, canning, heating, fermenting, chilling and freezing; some that affect the organoleptic properties of the product while others maintain the fresh taste of the food such as freezing and thawing (Alizadeh et al., 2007; Gonçalves, Nielsen & Jessen, 2012; Torry Research Station 1977). The primary gap identified by this review is much of the research into freezing and thawing methods is limited to novel methods (Cai et al., 2019; Li & Sun, 2002; Wu et al., 2017) or small-scale studies (Baygar & Alparslan, 2015; Cai et al., 2020; Espinoza Rodezno et al., 2013) with little research directed at fullscale, processing applications (Svendsen et al., 2022). Another gap identified is the use of uncontrolled thawing methods (Haugland, 2002) by industry and the relative lack of thawing research compared to freezing (Fig. 4).

Therefore, employing freezing and thawing throughout the value chain is an effective method to preserve seafood. As discussed, the GVC for demersal, pelagic and shellfish species is similar to freezing and thawing after primary processing, secondary processing and throughout the distribution chain to maintain freshness and quality. In higher lipid content pelagic species, freezing may be coupled with other preservation methods such as advanced packaging, salting and pickling to minimize oxidation. In mollusks, specifically bivalves, freezing alone is not an effective preservation technique and may be coupled with HPP and specialized packaging.

For the specific case of Northern cod with consideration to a re-emerging Newfoundland and Labrador cod fishery, the primary concerns identified were: (a) the seasonality of the fishery and its disconnect from peak demand, (b) distance between the harvesting and processing locations and potential markets, (c) potential weather disruptions affecting the transport of the final product and (d) Newfoundland and Labrador’s ability to penetrate the high-end market by providing a product that closely resembles fresh. Optimized freezing and thawing methods allow processors to (a) process, freeze, and cold store cod products during the fishing season to be sold when demand and value are higher; (b) ship the frozen product to a distribution center near major markets using cheaper refrigerated ground transportation; (c) thaw fillets to meet the demand using an optimized process; and (d) sell product to high-end restaurants and seafood shops as a chilled product, which will command a price closer to fresh. These markets can be attained by (a) employing high-speed freezing techniques that minimize ice crystal size and subsequent damage to the flesh combined with ultralow temperature cold storage and transportation to maintain product quality until needed; and (b) optimizing the cold chain with thawing methods that thaw the product quickly while minimizing dehydration, damage to cells and subsequent drip loss (Persson & Löndahl, 1993; Seremeti, 2007; Suh et al., 2017). Implementing the above practices will allow processors to produce a chilled product that closely resembles a fresh fillet and can be sold to high-end restaurants and retailers and enable the local seafood industry to earn more value for a frozen-thawed cod product by commanding prices similar to fresh fillets.

Supplemental Information

Supplemental Information 1 Post-mortem nucleotide catabolism sequence in fish

Source : (Huss et al., 1995; Surette, Gill & LeBlanc, 1988)

Click here for additional data file.

Supplemental Information 2 Typical freezing process

T0 - initial product temperature, TF –freezing temperature, TN –nucleation temperature, ΔT –supercooling temperature

Source: (Otero et al., 2016)

Click here for additional data file.

Supplemental Information 3 Data to produce Fig. 4

Click here for additional data file.

Supplemental Information 4 R-Code for Generating Fig. 8

Click here for additional data file.

The authors would like to thank Mr. Robert Verge and Dr. Leonard Lye for their valuable suggestions during the preparation of this manuscript.

Additional Information and Declarations

Competing Interests

Author Contributions

Data Availability

The authors declare there are no competing interests.

Pete Brown conceived and designed the experiments, performed the experiments, analyzed the data, prepared figures and/or tables, authored or reviewed drafts of the paper, and approved the final draft.

Deepika Dave conceived and designed the experiments, performed the experiments, authored or reviewed drafts of the paper, and approved the final draft.

The following information was supplied regarding data availability:

This is the R code used to generate Fig. 8.

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
