# Peer review of "Current freezing and thawing scenarios employed by North Atlantic fisheries: their potential role in Newfoundland and Labrador’s northern cod (Gadus morhua) fishery"

_PeerJ, doi:10.7717/peerj.12526_

## Round 0.1 · original submission · Minor Revisions

Reviewers have now commented on your paper. You will see that one of them is advising that you revise your manuscript. If you are prepared to undertake the work required, I would be pleased to reconsider my decision.

Reviewer 1 ·

Basic reporting

The contents of the article is clear and unambiguous, but the overall the language phrasing is more of academic nature. Although the article confirms to the scope of the journal, there is space for improvement with more broad and cross disciplinary interest. Introduction adequately introduce the subject in more of an academic pattern. Author can take up an analytical approach to the past and current researches based on the particular subject (eg.. broad analysis on studies under a different freezing methods). Overall the idea and suggestions were appreciable and will help to satisfy the cross disciplinary readers.

Experimental design

Article satisfy the scope of the journal. The review is organized logically with sufficient subsections. The first part of the review focused more on theoretical aspects and need more cross referencing in each subsection.

Validity of the findings

Conclusion identify the unresolved problems and suggestions for improving the current gaps in the value chain with reference to the emerging Newfoundland and Labrador's emerging cod fisheries.

Additional comments

Line 93 “To the best of our knowledge, this paper is the first comprehensive review of this broad scope.” Justify ?….Give focus on North Atlantic fisheries
Line 265 to 282 the word “Freezing” has been used in multiple times
Certain references were too old (eg: line no 405) can be revised
Line 508 –line 514: reframe the sentences to give more focus to the principle rather than specification of the equipment mentioned
Line 537-552 Authors can cite more references in the emerging high pressure thawing technologies
Line 615-616 Reframe the sentences in a professional language
Line 626- It is advisable to cite recent references in the article

·

Basic reporting

I find the article interesting, and it gives a good overview of the industrial technology status within freezing and thawing. Also, nice to learn more about Newfoundland and Labrador challenges. Nearly all the figures and all the tables seem adequate. I only have one comment regarding the manuscript.
• In line 314 there is a reference to Fig.2 that I don’t understand. Fig.2 shows an air blast freezer and a spiral freezer, both depend on air as cooling medium(direct) and not indirect.

Experimental design

The introduction present the content and purpose of the review in an structured manner. The sources are cited correct, and the structure of the paper is logically.

Validity of the findings

no comment

---

## Round 0.2 · accepted · Accept

I have completed my evaluation of your manuscript and it gives me great pleasure to inform you that your manuscript is now accepted for publication. Congratulations!